# STANDARD M10 SARS-CoV-2 Assay for Rapid Detection of SARS-CoV-2: Comparison of Four Real-Time PCR Assays

**DOI:** 10.3390/diagnostics12081998

**Published:** 2022-08-18

**Authors:** Seri Jeong, Nuri Lee, Su Kyung Lee, Eun-Jung Cho, Jungwon Hyun, Min-Jeong Park, Wonkeun Song, Hyun Soo Kim

**Affiliations:** 1Department of Laboratory Medicine, Hallym University Kangnam Sacred Heart Hospital, Hallym University College of Medicine, 1, Singil-ro, Yeongdeungpo-gu, Seoul 07441, Korea; 2Department of Laboratory Medicine, Hallym University Dongtan Sacred Heart Hospital, Hallym University College of Medicine, 7, Keunjaebong-gil, Hwaseong 18450, Korea

**Keywords:** SARS-CoV-2, real-time PCR, comparison, variant

## Abstract

The demand for assays that can rapidly and accurately detect severe acute respiratory syndrome coronavirus 2 (SARS-CoV-2) remains high. We evaluated the performance of two rapid real-time reverse transcription polymerase chain reaction (RT-qPCR) assays (STANDARD M10 SARS-CoV-2 and Xpert Xpress SARS-CoV-2) against conventional RT-qPCR assays (STANDARD M nCoV and Allplex SARS-CoV-2) for detecting SARS-CoV-2. A total of 225 swab samples were collected and tested using the four assays. The STANDARD M10 SARS-CoV-2 assay showed 97.4% positive percent agreement (PPA) and 100.0% negative percent agreement (NPA) compared to the STANDARD M nCoV assay and Allplex SARS-CoV-2 assay. STANDARD M10 exhibited high performance except in samples with low viral loads (cycle threshold (Ct) > 30). Xpert Xpress showed PPA and NPA of 100.0% compared to the two conventional RT-qPCR assays. The kappa coefficient (Κ) showed nearly almost perfect agreement between each assay and conventional RT-qPCR assays. The correlations of Ct values between the two rapid RT-qPCR and conventional RT-qPCR assays were >0.8, indicating strong correlations. All included assays could detect SARS-CoV-2 variants, such as the Alpha, Beta, and Gamma variants. The recently developed STANDARD M10 has a shorter turnaround time and random-access detection on automated devices, thereby facilitating efficient testing in emergency settings.

## 1. Introduction

Coronavirus disease 2019 (COVID-19), caused by severe acute respiratory syndrome coronavirus 2 (SARS-CoV-2), has spread globally. This pandemic has persisted for over 2 years, during which time numerous variants of concern have emerged [1,2]. As of April 1, 2022, 6142,735 cumulative deaths have occurred worldwide [3], greatly impacting social and healthcare systems. South Korea reported 14,001,406 cumulative confirmed cases and 17,453 deaths as of April 2022 with variant outbreaks [4]. Therefore, the demand for assays that can rapidly and accurately detect SARS-CoV-2 remains high.

Real-time reverse transcription polymerase chain reaction (RT-qPCR) assays are the standard method for detecting SARS-CoV-2 [5]. These procedures involve RNA extraction from the virus, reverse transcription into cDNA, amplification of target genes, and detection of fluorescent signals. This method enables sensitive detection in samples with low viral loads through amplification and detection of viral nucleic acids. However, RT-qPCR assays require skilled laboratory personnel, infrastructure, and a long turnaround time. Furthermore, the assay is costly, and reagent and material shortages have occurred during the COVID-19 pandemic [6]. 

Automated devices that can test for SARS-CoV-2 within 1 h have been developed [7,8]. The process from sample loading to result interpretation is entirely automated. These methods do not always require skilled laboratory personnel or specialized infrastructure, enabling their use at point-of-care facilities. Although their main limitation is reduced sensitivity compared to that of a conventional RT-qPCR test, random access and rapid detection are advantageous in emergency settings.

Numerous evolving SARS-CoV-2 variants have rapidly emerged worldwide. The World Health Organization has assigned Greek letters to name notable variants [9]. The Centers for Disease Control and Prevention identified the first four variants of concern circulating globally as Alpha (B.1.1.7 and Q lineage), Beta (B.1.351 and descendent lineages), Gamma (P.1 and descendent lineages), and Delta (B.1.617.2 and AY lineages) variants, which are more transmissible than wild-type SARS-CoV-2 strains and contain specific protein mutations (https://www.cdc.gov/coronavirus/2019-ncov/variants/variant-classifications.html, accessed on 17 May 2022). The Alpha variant was first identified in the United Kingdom in late 2020 and became dominant worldwide before the emergence of the Delta variant [10,11,12]. The Beta variant predominated in South Africa in late 2020, and the Gamma variant was prevalent in Brazil in late 2020 [13,14]. The Omicron variant was first reported in Botswana and South Africa in November 2021 and has persisted until this study period. This variant harbors over 30 mutations in the spike protein that enhance transmissibility and reduces susceptibility to neutralizing antibodies [15]. Therefore, the Omicron variant has contributed to the largest number of SARS-CoV-2 infections in many countries, posing a severe threat to global health care systems [16,17].

The STANDARD M10 SARS-CoV-2 assay was recently developed as a rapid RT-qPCR assay. In this study, the performance of this method was compared to that of Xpert Xpress SARS-CoV-2, which can rapidly and accurately detect SARS-CoV-2 [7,18], and with that of conventional RT-qPCR assays, such as the STANDARD M nCoV real-time-detection kit and Allplex SARS-CoV-2 assay [19,20]. In addition, the ability to detect previously prevalent SARS-CoV-2 variants of concern was evaluated using reference materials [21].

## 2. Materials and Methods

### 2.1. Study Population and Sample Collection 

A total of 225 nasopharyngeal swabs were collected from patients for testing with SARS-CoV-2 RT-qPCR tests from 14 January to 22 April 2022. In Korea, the Omicron variant has been predominantly detected since 20 January 2022 after it was first isolated on 1 December 2021. The patients comprised 124 men and 101 women. The median age of the study population was 41.0 years (range: 0–99 years). The swab samples were stored in viral transport medium (Clinical Virus Transport Medium, Noble Bio, Hwaseong, Korea) and tested using two rapid nucleic acid amplification assays, which can provide results within 1 h. The results were compared with those obtained using two RT-qPCR assays that are widely used as references for detecting SARS-CoV-2. Samples without the required volume and those duplicated from a single patient were excluded. Nine samples with inconclusive results showing partial positivity among multiple target genes in each assay (Appendix A) were not included in the analysis [22,23]. One sample with an invalid result was also excluded. Overall, 215 patient samples were analyzed. In addition, the AccuPlex SARS-CoV-2 Variant Panel 1 (SeraCare Life Sciences, Milford, MA, USA) was used as reference material to evaluate the detection capacity for important SARS-CoV-2 variants. The panel consisted of five vials: B.1.1.7 (Alpha) variant vial, B.1.351 (Beta) variant vial, P.1 (Gamma) variant vial, wild-type vial (NCBI Reference sequence NC_045512.2, Wuhan-Hu-1), and RNase P vial as a negative control.

### 2.2. STANDARD M10 SARS-CoV-2 as a Rapid RT-qPCR Assay

The STANDARD M10 SARS-CoV-2 (SD Biosensor, Suwon, Korea) test is an automated in vitro diagnostic RT-qPCR assay for qualitative detection of SARS-CoV-2 within 1 h. The STANDARD M10 SARS-CoV-2 test was performed using the STANDARD M10 system, which integrates sample preparation, nucleic acid extraction, amplification, and detection of target sequences in nasopharyngeal specimens using molecular diagnostic assays. The system consists of a STANDARD M10 Module and STANDARD M10 Console with preloaded software for analyzing the results. The system requires the use of disposable cartridges that contain the RT-qPCR reagents during processing. Cross-contamination between samples was minimized because the cartridges are self-contained. Pneumatic pressure was utilized to transfer the samples and fluids through the chamber in the cartridge. A total of 600 μL of viral transport medium was required. The STANDARD M10 SARS-CoV-2 assay targets the envelope (E) gene and open reading frame 1ab (ORF1ab) gene. After 40 RT-qPCR cycles, positive results were obtained if the cycle thresholds (Ct) of the two genes were within 35.0 cycles. All experiments were performed in accordance with the manufacturer’s instructions. Photographs of all included assays are presented in Appendix A.

### 2.3. Xpert Xpress SARS-CoV-2 as a Rapid RT-qPCR Assay

Xpert Xpress SARS-CoV-2 (Cepheid, Sunnyvale, CA, USA) is an automated real-time RT-qPCR test carried out in single-use disposable cartridges. This assay requires 1 h (from sample loading to verifying the results after analysis). For this assay, 300 μL of viral transport medium was loaded into the sample chamber of the cartridge using a sterile pipette. The cartridges were placed in a GeneXpert System (Cepheid) and processed. This assay targets the E gene and nucleocapsid (N) gene. If both genes showed Ct values within the 40 cycles of the RT-qPCR, the result was considered as positive for the detection of SARS-CoV-2.

### 2.4. Nucleic Acid Extraction for Conventional RT-qPCR Assay

Nucleic acid extraction for conventional RT-qPCR assays such as the STANDARD M nCoV and Allplex SARS-CoV-2 assays was performed. A QIAamp Viral RNA Mini Kit (Qiagen, Hilden, Germany) and the QIAcube platform were used to isolate nucleic acids for detecting SARS-CoV-2 using RT-qPCR assays that are widely used as references. All procedures were performed according to the manufacturer’s instructions.

### 2.5. STANDARD M nCoV Real-time Detection Kit as a Conventional RT-qPCR Assay

The STANDARD M nCoV real-time detection kit (SD Biosensor) is based on the TaqMan probe real-time fluorescent qPCR aimed at qualitative detection of SARS-CoV-2 nucleic acids from nasopharyngeal and oropharyngeal swabs in viral transport medium. This assay requires 10 μL of specimen and 90 min of run time. Amplification and detection were performed on a Bio-Rad CFX96 thermocycler (BioRad Laboratories, Hercules, CA, USA). The FAM, JOE, and CY5 channels were used for qualitative detection of the ORF1ab gene, E gene, and internal control, respectively. To prevent contamination of the amplification products, dUTP and uracil DNA glycosylase enzymes were used in this assay. The lower detection limit was 0.5 copies/μL. When the Ct values of the ORF1ab and E gene were within the cutoff of 36.0, the result was considered to be positive.

### 2.6. Allplex SARS-CoV-2 Assay as a Conventional RT-qPCR Assay

The Allplex SARS-CoV-2 assay (Seegene, Seoul, Korea) is a real-time RT-qPCR assay for detecting and identifying three target genes (E, N, and RNA-dependent RNA polymerase (RdRP)) in SARS-CoV-2 in a single tube. Amplification and detection were performed for 45 cycles on a Bio-Rad CFX96 thermocycler (BioRad Laboratories). The running time was 110 min after extraction. The results were interpreted using Seegene Viewer data analysis software. The Ct was automatically determined using the manufacturer’s software. The cutoff in this assay was 40. The result was considered as positive when the Ct values of all these three genes were within this cutoff.

### 2.7. Statistical Analysis 

Analyse-it Method Evaluation Edition software, version 2.26 (Analyse-it Software Ltd., Leeds, UK) was used to evaluate descriptive statistics. Agreement levels between assays were determined based on Cohen’s kappa coefficient values. Agreement was categorized as follows: slight, 0.00–0.20; fair, 0.21–0.40; moderate, 0.41–0.60; substantial, 0.61–0.80; and almost perfect, 0.81–1.00. Spearman’s rank correlation coefficients were used to examine the correlations between rapid RT-qPCR assays and conventional RT-qPCR assays. Values between 0.70 and 0.89 were considered to be strong, and those above 0.90 were interpreted as very strong. MedCalc software, version 19.8 (MedCalc Software Ltd., Ostend, Belgium) was used for correlation analysis. 

## 3. Results

### 3.1. Performance of STANDARD M10 and Xpert Xpress SARS-CoV-2 Assays

The final 215 swabs comprised 114 positive and 101 negative samples (determined using two conventional RT-qPCR assays). Using these samples, the performance of the STANDARD M10 SARS-CoV-2 and Xpert Xpress SARS-CoV-2 assays was assessed. When the results of the STANDARD M nCoV and Allplex SARS-CoV-2 assays were used as references, the positive percent agreement (PPA), negative percent agreement (NPA), and total agreement of the STANDARD M10 SARS-CoV-2 assay were 97.4%, 100%, and 98.6%, respectively (Table 1). The PPA, NPA, and total agreement of the Xpert Xpress SARS-CoV-2 assay were all 100.0%. When using samples with Ct values < 30 in the two conventional RT-qPCR assays, the PPA of the STANDARD M10 SARS-CoV-2 and Xpert Xpress SARS-CoV-2 was 100.0% (Table 2). However, when using 24 samples in the Allplex 2019 nCoV assay with Ct values > 30, the PPA of STANDARD M10 compared to Allplex 2019 nCoV decreased to 87.5% (with 3 of 24 samples showing discrepant results). When using five samples in the STANDARD M nCoV assay with Ct values > 30, the PPA of STANDARD M10 compared to STANDARD M nCoV was 40.0% (with 3 of 5 samples showing discrepant results). The Ct values of three samples with discrepant results were 34.7, 36.5, and 37.2 for the Allplex SARS-CoV-2 E gene and 32.0, 32.8, and 34.3 for the STANDARD M nCoV E gene.

### 3.2. Discordant Results

Among the 114 swabs determined to be positive using conventional RT-qPCR assays, three samples had Ct values >30 and showed discrepant results compared to the results obtained using other assays (Table 3). The Ct values obtained using Allplex SARS-CoV-2 were 36.5, 37.0, and 36.4 for the E, RdRP, and N genes, respectively. The median Ct values measured by STANDARD M nCoV were 32.8 and 32.2 for the E and ORF1ab genes, respectively. The Ct values obtained using Xpert Xpress were 37.4 and 38.4 for the E and N genes, respectively, indicating low viral loads.

### 3.3. Correlation between SARS-CoV-2 Assays

The correlations between the performance of the two rapid RT-qPCR tests and two conventional RT-qPCR assays were analyzed (Figure 1). The Ct values for the E gene were used, as this gene is detected in all included assays. Spearman’s rank correlation coefficients for the correlations between the rapid RT-qPCR assays and conventional RT-qPCR assays were 0.80–0.90 (Allplex SARS-CoV-2 vs. STANDARD M10, 0.877; Allplex SARS-CoV-2 vs. Xpert Xpress SARS-CoV-2, 0.879; STANDARD M nCoV vs. STANDARD M10, 0.898; and STANDARD M nCoV vs. Xpert, 0.887), indicating strong correlations based on predefined criteria. 

### 3.4. Results for Variant Reference Materials

All included rapid and conventional RT-qPCR assays detected the AccuPlex SARS-CoV-2 variant reference materials, demonstrating effective performance with respect to detecting SARS-CoV-2 variants. The Ct values measured for variants B.1.1.7 (Alpha), B.1.351 (Beta), and P.1 (Gamma) as well as the wild-type virus are presented in Table 4.

## 4. Discussion

We investigated the performance of two rapid RT-qPCR tests compared to those of conventional RT-qPCR assays. The PPA and NPA values of STANDARD M10 and Xpert Xpress (compared to conventional RT-qPCR tests) were over 95%. The agreement between assays was near perfect based on the kappa values. Discrepant results were observed in samples with low viral loads. Variants of SARS-CoV-2 were detected in all assays with strong agreement according to Spearman’s rank correlation coefficients. All included rapid and conventional RT-qPCR assays detected variants of concern including Alpha, Beta, and Gamma.

Many molecular diagnostic platforms with sample-to-answer settings for the detection of SARS-CoV-2 have been developed and evaluated [6,7,24]. Although conventional RT-qPCR assays are used as references, they require skilled personnel, infrastructure, and time-consuming processes, including sample collection and analysis in batches. However, these rapid RT-qPCR devices enable random access, which is useful for emergency settings. Furthermore, these assays are simple and have short turnaround times. Xpert Xpress, included in this study, is a rapid RT-qPCR assay that has been widely used and evaluated [8,18,22,24,25,26]. This assay requires 45 min from sample to answer. The turnaround time can be reduced if the amplification curve reaches the threshold earlier. According to a meta-analysis of 11 studies evaluating the Xpert Xpress assay [7], the pooled sensitivity was 99.0% and specificity was 97.0%, similar to our results. The agreement between Xpert Xpress and a conventional RT-qPCR assay was 0.99 in a previous study [22], indicating near-perfect agreement [27]. Additionally, a very strong correlation between Xpert Xpress and a conventional RT-qPCR assay (R^2^ = 0.94) was recently observed [22]. 

STANDARD M10 was also developed as a rapid RT-qPCR assay and showed results comparable with those obtained using conventional methods (PPA, 97.4%; NPA, 100.0%; and kappa, 0.97). The PPA of STANDARD M10 was reduced when the Ct values exceeded 30 with the conventional RT-qPCR test. According to previous studies, other rapid RT-qPCR assays showed decreased sensitivity in samples with low viral loads [22,28]. The rapid molecular assays Xpert Xpress and ID NOW were compared with the Roche Cobas SARS-CoV-2 assay as a reference for samples with diverse SARS-CoV-2 viral concentrations [28]. When the Ct values were <30, indicating medium and high viral concentrations, both Xpert Xpress and ID Now showed 100% PPA. For samples with Ct values > 30, the PPA of Xpert Xpress was 97.1%, whereas that of ID NOW was 34.3%, indicating a limited ability to evaluate samples with low viral concentrations. Another study demonstrated that respiratory samples with low viral loads were likely to yield false negative results in molecular diagnostic platforms with sample-to-answer settings [29]. The sample-to-answer setting is the seamless process enabling the rapid return of results after adding the samples to the dedicated reagent [22]. Furthermore, a short article reported that all false negative results stem from samples with low viral concentrations (Ct values of 35–40) [30]. 

The Allplex 2019-nCoV assay and STANDARD M nCoV used as references are commercially available real-time RT-qPCR reagents approved by the Korea Centers for Disease Control and Prevention [23]. These assays have been evaluated and showed reliable performance in previous studies [19,20,31]. We adopted two widely used reagents because subtle differences can occur in each assay. The target genes and primer-probe sets used for the conventional RT-qPCR assays can influence their analytical sensitivity and efficiency [5,32]. To determine the influence of these factors on the results of the RT-qPCR assays used as references, we performed the tests using two assays. In addition, differences in the sampling day and testing day may affect the results. According to the guidelines for laboratory diagnosis of COVID-19 in South Korea [23], storage within 5 days at 4℃ is recommended for upper respiratory tract specimens including nasopharyngeal swabs. Therefore, to minimize differences arising from sampling and storage, we conducted two rapid RT-qPCR assays and nucleic acid extractions for conventional RT-qPCR assays immediately after obtaining the samples. Furthermore, the interpretation criteria can influence the results. Detection of only one of the multiple genes can be interpreted as positive for COVID-19 based on instructions from certain manufacturers. However, the Korean Society for Laboratory Medicine recommends the determination of a positive result only when all genes are detected considering the data from many COVID-19 cases [23]. When only one gene is identified, resampling and retesting or consultation of the reference laboratory is recommended. Therefore, we considered a result as positive only when the Ct values of all genes were within the cutoff provided by the manufacturer to ensure that our data were robust based on the guidelines for laboratory diagnosis of COVID-19 provided by the Korean Society for Laboratory Medicine [23]. Partially positive results are shown in Appendix A; these samples had low viral titers. 

SARS-CoV-2 has mutated over time, leading to genetic variations in the population of circulating viral strains during the COVID-19 pandemic. The increased transmission of variants, which were derived from the United Kingdom (B.1.1.7/Alpha), South Africa (B1.351/Beta), and Brazil (P.1/Gamma), requires a rapid public health response [33]. Molecular tests are influenced by these variants because of inherent design differences in each assay [34]. Viral mutations have been suggested to impact the performance of the Xpert Xpress assay [34,35]. Mutations within the regions targeted by the assay may negatively affect primer or probe binding, leading to failure to detect the presence of SARS-CoV-2. Studies of the performance of rapid and conventional RT-qPCR assays against these variants remain limited. We found that the STANDARD M10 and Xpert Xpress as well as STANDARD M nCoV and Allplex 2019-nCoV assays could detect variants of concern, including Alpha, Beta, and Gamma.

This study presented certain limitations. We included positive samples only, i.e., those with Ct values within the predefined cutoff for all genes targeted in the RT-qPCR assays. Samples with positivity for only a subset of the chosen genes were not included, raising concerns regarding the performance of the assays. However, our criteria for positivity were based on the guidelines recommended by the Korean Society for Laboratory Medicine [23]. In addition, sample testing delays because of holidays can affect the results, despite the guideline enabling up to 5-day storage for upper respiratory tract samples. Furthermore, estimation of positive or negative predictive values was not possible because samples were not collected consecutively, and the prevalence could not be determined. Further studies of a larger number of samples with more diverse SARS-CoV-2 variants are necessary to validate the performance of these assays.

## 5. Conclusions

STANDARD M10 and Xpert Xpress developed as rapid RT-qPCR assays showed high positive and negative agreement when the results of the conventional RT-qPCR assay (Allplex 2019-nCoV and STANDARD M nCoV) were used as a reference. STANDARD M10 and Xpert Xpress also showed near-perfect agreement and a strong correlation with conventional RT-qPCR assays, indicating their nearly equivalent test accuracy. STANDARD M10 was less effective for samples with low viral loads. However, this assay requires minimal technical skills and enables random access and rapid return of the test results, similar to Xpert Xpress. In addition, all included assays detected SARS-CoV-2 variants, which have caused serious infection outbreaks. The performance of STANDARD M10, STANDARD M nCoV, and Allplex 2019-nCoV against the variants has not been widely reported. Therefore, utilizing these RT-qPCR assays can improve the turnaround time and efficiency of the testing system in hospitals.

## Figures and Tables

**Figure 1 diagnostics-12-01998-f001:**
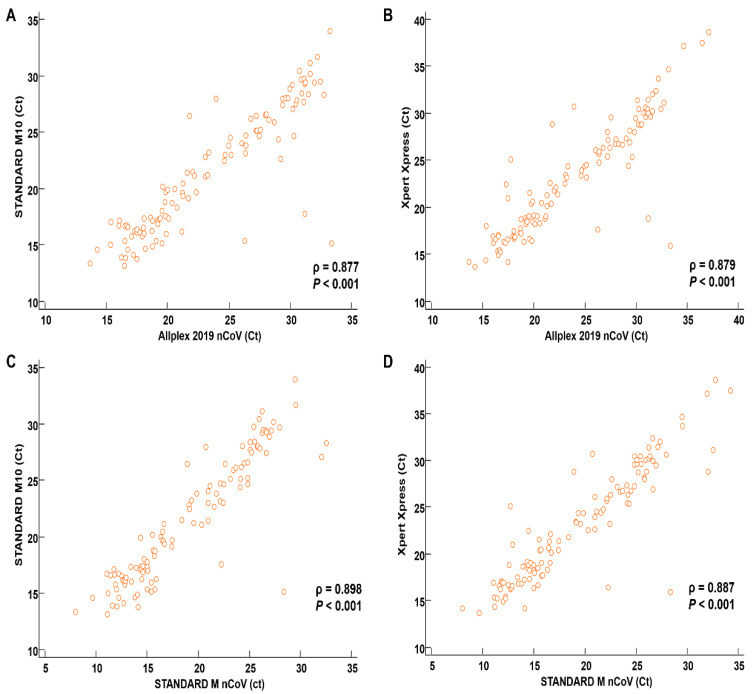
Correlations between E gene Ct values of two rapid RT-qPCR assays and conventional RT-qPCR assays. (**A**) Allplex SARS-CoV-2 vs. STANDARD M10 SARS-CoV-2; (**B**) Allplex SARS-CoV-2 vs. Xpert Xpress SARS-CoV-2; (**C**) STANDARD M nCoV vs. STANDARD M10 SARS-CoV-2; (**D**) STANDARD M nCoV vs. Xpert Xpress SARS-CoV-2. Ct, cycle threshold.

**Table 1 diagnostics-12-01998-t001:** Positive percent agreement and negative percent agreement of STANDARD M10 SARS-CoV-2 assay and Xpert Xpress SARS-CoV-2 assay compared to two conventional RT-qPCR assays.

Performance ^1^	Allplex 2019-nCoV as a Reference		STANDARD M nCoV as a Reference	
	STANDARD M10 SARS-CoV-2	Xpert Xpress SARS-CoV-2	STANDARD M10 SARS-CoV-2	Xpert Xpress SARS-CoV-2
P/P (*n*)	111	114	111	114
P/N (*n*)	3	0	3	0
N/P (*n*)	0	0	0	0
N/N (*n*)	101	101	101	101
PPA (%) ^1^	97.4 (92.5–99.5)	100.0 (96.8–100.0)	97.4 (92.5–99.5)	100.0 (96.8–100.0)
NPA (%) ^1^	100.0 (96.4–100.0)	100.0 (96.4–100.0)	100.0 (96.4–100.0)	100.0 (96.4–100.0)
Total agreement (%)^1^	98.6 (96.0–99.7)	100.0 (98.3–100.0)	98.6 (96.0–99.7)	100.0 (98.3–100.0)
Kappa value^1^	0.97 (0.94–1.00)	1.00 (1.00–1.00)	0.97 (0.94–1.00)	1.00 (1.00–1.00)

^1^ Values are expressed as number or % (95% confidence interval). Target genes in each assay: E and ORF1ab genes for STANDARD M10 SARS-CoV-2 and STANDARD M nCoV; E and N2 genes for Xpert Xpress SARS-CoV-2; E, RdRP, and N genes for Allplex 2019-nCoV. P, positive; N, negative; PPA, positive percent agreement; n, number; NPA, negative percent agreement. Samples with inconclusive results showing partial positivity among multiple target genes for each SARS-CoV-2 assays are excluded from this table and are shown in Appendix A.

**Table 2 diagnostics-12-01998-t002:** Positive percent agreement of STANDARD M10 and Xpert Xpress depending on two RT-qPCR Ct values.

Ct Value	Allplex 2019-nCoV as a Reference		STANDARD M nCOV RT-qPCR as a Reference	
	STANDARD M10 SARS-CoV-2	Xpert Xpress SARS-CoV-2	STANDARD M10 SARS-CoV-2	Xpert Xpress SARS-CoV-2
<20	100.0% (*n* = 42)	100.0% (*n* = 42)	100.0% (*n* = 62)	100.0% (*n* = 62)
20–25	100.0% (*n* = 25)	100.0% (*n* = 25)	100.0% (*n* = 28)	100.0% (*n* = 28)
26–30	100.0% (*n* = 23)	100.0% (*n* = 22)	100.0% (*n* = 19)	100.0% (*n* = 19)
>30	87.5% (*n* = 24)	100.0% (*n* = 24)	40.0% (*n* = 5)	100.0% (*n* = 5)

Samples with inconclusive results showing partial positivity among multiple target genes for each SARS-CoV-2 assay are excluded from this table and are shown in Appendix A. Ct, cycle threshold.

**Table 3 diagnostics-12-01998-t003:** Ct values of three discrepant results showing negative only in STANDARD M10.

Each Assay (Target Gene)	Ct Values
Allplex 2019-nCoV (E)	36.5 (35.0–37.1)
Allplex 2019-nCoV (RdRP)	37.0 (35.0–37.2)
Allplex 2019-nCoV (N)	36.4 (34.7–36.8)
STANDARD M nCoV (E)	32.8 (32.1–34.1)
STANDARD M nCoV (ORF1ab)	32.2 (31.0–33.8)
Xpert Xpress (E)	37.4 (37.2–38.4)
Xpert Xpress (N2)	38.4 (36.8–39.0)

Quantitative values are expressed as median (95% confidence interval). Ct, cycle threshold.

**Table 4 diagnostics-12-01998-t004:** Ct values of each gene of SARS-CoV-2 assay obtained using the AccuPlex SARS-CoV-2 variant Panel 1.

Target Gene in Each Assay	B.1.1.7 (Alpha)	B.1.351 (Beta)	P.1 (Gamma)	Wild-Type
Allplex 2019-nCoV (E)	31.22	30.47	31.95	32.62
Allplex 2019-nCoV (RdRP)	32.79	30.39	31.77	31.69
Allplex 2019-nCoV (N)	32.13	31.83	33.11	33.6
STANDARD M nCoV (E)	28.19	28.19	29.48	29.26
STANDARD M nCoV (ORF1ab)	27.92	28.03	29.12	29.25
STANDARD M10 (E)	27.89	28.31	28.49	28.94
STANDARD M10 (ORF1ab)	27.6	27.99	28.21	28.08
Xpert Xpress SARS-CoV-2 (E)	30.1	30	30.7	31.1
Xpert Xpress SARS-CoV-2 (N2)	33.2	32.4	34.1	34

Ct, cycle threshold.

## Data Availability

The data used and presented in this study were deposited in https://dataverse.harvard.edu/. (https://doi.org/10.7910/DVN/VRRHFU).

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
