# Peer review of "STANDARD M10 SARS-CoV-2 Assay for Rapid Detection of SARS-CoV-2: Comparison of Four Real-Time PCR Assays"

_diagnostics, 2022, doi:10.3390/diagnostics12081998_

Round 1
Reviewer 1 Report
Dear editor
The authors evaluated the performance of two rapid real-time polymerase chain reaction (qPCR) assays and make a comparison with Xpert Xpress. The study was well designed and the results were convincing, however, the development and evaluation of such test are not novel, and the test is still time-consuming regarding both Nucleic Acid Extraction and amplication. In our lab, both Nucleic Acid Extraction and application would be completed in 2h and the amplication process would be completed within 1 h, two years ago.
Author Response
Response to Reviewer 1 Comments
The authors evaluated the performance of two rapid real-time polymerase chain reaction (qPCR) assays and make a comparison with Xpert Xpress. The study was well designed and the results were convincing, however, the development and evaluation of such test are not novel, and the test is still time-consuming regarding both Nucleic Acid Extraction and amplication. In our lab, both Nucleic Acid Extraction and application would be completed in 2h and the amplification process would be completed within 1 h, two years ago.
Response: We described the newly developed STANDARD M10, which requires only 1 h for all procedures, including nucleic acid extraction and amplification. This test takes only 59 min from sample loading to the detection of SARS-CoV-2. Thus, the turnaround time of this assay is shorter (59 min) than that of other conventional real-time PCR assays and Xpert Xpress (70 min).
Reviewer 2 Report
Evaluation of STANDARD M10 SARS-CoV-2 Assay for the Rapid Detection of SARS-CoV-2: Comparison of Four Real-time PCR Assays
General comments
The study does well to address a relevant research gap regarding assays for detection of SARS-CoV-2. The study evaluates the performance of two assays (STANDARD M10 assay and Xpert Xpress assay) for SARS-CoV-2 detection comparing it performance with other assays and is found be effective for facilitating efficient testing. The study is well designed with nice flow of concepts. However, the weakness in the study lies in not focusing on evaluating the assays on the omicron variant, the currently predominating global variant although the authors have indicated that studies on the omicron variant remain scarce.
Specific comments
Abstract & Introduction
The abstract and introduction is well written and relevant.
Materials and Methods
Study population and sample collection
- Provide manufacturer name/details of the viral transport medium and also the RNase P.
STANDARD M10 as a Rapid qPCR assay
-Spelling error: STANDARD not STANDARD – for the sentence beginning with STANDARD M10 targets…
Results, Discussion and Conclusion
These sections are very well presented.
The authors mainly only report UK B.1.1.7, South African B.1.351 and Brazil P.1. Considering the current medical importance of the omicron variant, why did the authors not try to evaluate the performance on this highly transmittable variant that has predominated over other variants?
Author Response
General comments
The study does well to address a relevant research gap regarding assays for detection of SARS-CoV-2. The study evaluates the performance of two assays (STANDARD M10 assay and Xpert Xpress assay) for SARS-CoV-2 detection comparing it performance with other assays and is found be effective for facilitating efficient testing. The study is well designed with nice flow of concepts. However, the weakness in the study lies in not focusing on evaluating the assays on the omicron variant, the currently predominating global variant although the authors have indicated that studies on the omicron variant remain scarce.
Response: We appreciate your positive comments. In this study, the sample collection period was from January 14 to April 22, 2022, when the Omicron variant was predominantly spreading across South Korea. Therefore, we did not include the analysis of Omicron variants because the patients’ samples presented mostly Omicron and some Delta variants.
We have added a sentence regarding the Omicron variant to the revised Materials and Methods section (page 3, lines 77–79).
“A total of 225 nasopharyngeal swabs were collected from patients for SARS-CoV-2 RT-qPCR tests from January 14 to April 22, 2022. In Korea, the Omicron variant has been predominantly detected since January 20, 2022 after it was first isolated in December 1, 2021.”
Specific comments
Abstract & Introduction
The abstract and introduction is well written and relevant.
Materials and Methods
Study population and sample collection
Provide manufacturer name/details of the viral transport medium and also the RNase P.
Response: We have added the manufacturer name/details of the viral transport medium to the revised Materials and Methods section. RNase P vial was included as a negative control in the AccuPlex SARS-CoV-2 Variant Panel 1.
Materials and Methods (page 2, lines 81–82)
“The swab samples were stored in viral transport medium (Clinical Virus Transport Medium, Noble Bio, Hwaseong, Korea) and tested using two rapid nucleic acid amplification assays.”
Materials and Methods (page 2, lines 89–94)
“In addition, the AccuPlex SARS-CoV-2 Variant Panel 1 (SeraCare Life Sciences, Milford, MA, USA) was used as reference material to evaluate the detection capacity of important SARS-CoV-2 variants. The panel consisted of five vials: B.1.1.7 (Alpha) variant vial, B.1.351 (Beta) variant vial, P.1 (Gamma) variant vial, Wild Type vial (NCBI Reference sequence NC_045512.2, Wuhan-Hu-1), and RNase P vial as a negative control.”
STANDARD M10 as a Rapid qPCR assay
-Spelling error: STANDARD not STANDARD – for the sentence beginning with STANDARD M10 targets…
Response: Thank you for your comment. This term has been corrected to “STANDARD”.
Results, Discussion and Conclusion
These sections are very well presented.
The authors mainly only report UK B.1.1.7, South African B.1.351, and Brazil P.1. Considering the current medical importance of the omicron variant, why did the authors not try to evaluate the performance on this highly transmittable variant that has predominated over other variants?
Response: In this study, the sample collection period was from January 14 to April 22, 2022, when the Omicron variant predominantly spread in South Korea. Therefore, we did not include analysis of Omicron variants because the patients’ samples were mostly Omicron and some Delta variants.
We have added a sentence regarding the Omicron variant to the revised Materials and Methods section (page 3, lines 77–79).
“A total of 225 nasopharyngeal swabs were collected from patients for SARS-CoV-2 RT-qPCR tests from January 14 to April 22, 2022. In Korea, the Omicron variant has been predominantly detected since January 20, 2022 after it was first isolated in December 1, 2021.”
Reviewer 3 Report
The manuscript is interesting and follows my comments:
You should change the names of the SARS-CoV-2 variants as describes at https://www.who.int/activities/tracking-SARS-CoV-2-variants/, such as alpha, beta, gamma and omicron.
Correction
1- severe acute respiratory syndrome coronavirus 2 (SARS-CoV-2) should be described only in the first sentece and only SARS-CoV-2 in the line 14 of the abstract;
2- in the line 7 of the introduction is in April or until April?
3- In the last sentence of the introduction I recomend to describe the variants did you test in the manuscript and which one is prodominat nowadays.
4- The sentence: Nine samples with incomplete results showing partial positivity among multiple genes for SARS-CoV-2 (supplementary Table S1) and one sample with an invalid result were also excluded " is not clear.
5- In section 2.5 and 2.3 will be nice to have a picture of the Xpert Xpress, Standard M10, Allplex 2019 - nCoV and Standard nCoV RT-qPRC kits to understand the procedure.
6- in the sentence: "The final 197 swabs comprised 114 positive and 101 negative samples "should be 215 insted of 197?
7- In the legend of table 1 I suggest to describe the genes detected in each test.
8- the sub-index 1 is not in the table, should be add in PPA and NAP;
9- In the sentence "Samples with 3 incomplete results showing partial positivity among multiple genes for SARS-CoV-2 4 were excluded from this table and these results were shown in supplementary Table S1. "I will include the amout of samples not used.
10- I got confused that in the introduction is describe 215 samples, but in tables are described 225 samples and are described that samples with incomplete results were excluded. I think this inssue must be clarify in the text. How many samples were tested? How many were actually used in the Tables for the calculations of PPA, NAP and Total agreement.
10- In the line 52 change Table 2 to Table 4;
11- The results described in the Table 4 are not described in materials and methods. The amount of samples tested is not clear in the Table 4.
12- In line 60 describe the variants tested in the manuscript.
13- Line 76 change 0.987 to 0.99;
14- Line 78 change 0.9352 to 0.94;
15-line 83: samples with low viral load (reference?). Is missing a reference;
16- lines 91-92 is confused: "molecular diagnostic platforms with sample-to-answer settings ". What is the meaning of sample-to-answer settings?
17- lines 97-98 is confused: We adopted two widely 97 used reagents because subtle differences could be generated based on the reference assay.
18- lines 114 is confused: within the cutoff for robust data
19- line 125, what is N2 gene?
20- line 127: variants of concern such as N501Y and E484K . N501Y and E484K are not variants but mutations found in the spike protein. This sentence must be corrected;
